# Fisetin and Quercetin: Promising Flavonoids with Chemopreventive Potential

**DOI:** 10.3390/biom9050174

**Published:** 2019-05-06

**Authors:** Dharambir Kashyap, Vivek Kumar Garg, Hardeep Singh Tuli, Mukerrem Betul Yerer, Katrin Sak, Anil Kumar Sharma, Manoj Kumar, Vaishali Aggarwal, Sardul Singh Sandhu

**Affiliations:** 1Department of Histopathology, Postgraduate Institute of Medical Education and Research (PGIMER), Chandigarh 160012, Punjab, India; make.must@gmail.com (D.K.); vaishali.pgi@gmail.com (V.A.); 2Department of Biochemistry, Government Medical College and Hospital (GMCH), Chandigarh 160031, Punjab, India; garg.vivek85@gmail.com; 3Department of Biotechnology, Maharishi Markandeshwar (Deemed to be University), Mullana-Ambala 133 207, Haryana, India; anibiotech18@gmail.com; 4Department of Pharmacology, Faculty of Pharmacy, Erciyes University, Kayseri 38039, Turkey; eczbetul@yahoo.com; 5NGO Praeventio, Tartu 50407, Estonia; katrin.sak.001@mail.ee; 6Department of Chemistry, Maharishi Markandeshwar University, Sadopur 134007, Haryana, India; Manojraju27@gmail.com; 7Department of Biological Sciences, RD University, Jabalpur 482001, India; sardulsinghsandhu@gmail.com

**Keywords:** apoptosis, cell cycle arrest, extracellular matrix remodeling, epithelial to mesenchymal transition, signaling cascades, flavonoids, fisetin, quercetin

## Abstract

Despite advancements in healthcare facilities for diagnosis and treatment, cancer remains the leading cause of death worldwide. As prevention is always better than cure, efficient strategies are needed in order to deal with the menace of cancer. The use of phytochemicals as adjuvant chemotherapeutic agents in heterogeneous human carcinomas like breast, colon, lung, ovary, and prostate cancers has shown an upward trend during the last decade or so. Flavonoids are well-known products of plant derivatives that are reportedly documented to be therapeutically active phytochemicals against many diseases encompassing malignancies, inflammatory disorders (cardiovascular disease, neurodegenerative disorder), and oxidative stress. The current review focuses on two key flavonols, fisetin and quercetin, known for their potential pharmacological relevance. Also, efforts have been made to bring together most of the concrete studies pertaining to the bioactive potential of fisetin and quercetin, especially in the modulation of a range of cancer signaling pathways. Further emphasis has also been made to highlight the molecular action of quercetin and fisetin so that one could explore cancer initiation pathways and progression, which could be helpful in designing effective treatment strategies.

## 1. Introduction

The incidence of malignant diseases and the prevalence of cancer mortality is proliferating at an amplified rate across the developed and developing countries [1]. New Globocan 2018 cancer data from 185 countries documented 18.1 million new cancer cases and 9.6 million cancer-related deaths (GALOBOCAN 2018). Although the improvement of diagnostic tools, advanced treatment approaches, and cancer awareness programs have resulted in a remarkable drop in cancer mortality in the United States, cancer prevalence is still growing continuously [1]. This is attributed to smoking habits, alcohol consumption, unhealthy food, stress, and no or insufficient exercise amongst people in the developed and developing world [2]. There are different therapeutic modalities for cancer treatment available, including surgery, radiation therapy, chemotherapy, immunotherapy, and targeted therapy. Tamoxifen in estrogen receptor-positive breast tumor, Herceptin in Her 2+ breast cancer, epithelial growth factor receptor (EGRF) inhibitors (erlotinib, afatinib, osimertinib, and fefitinib) in non-small cell lung carcinoma (NSCLC), *K-ras* inhibitors (cetuximab) in colon cancer, *B-Raf* inhibitors (vemurafenib, dabrafenib, and encorafenib) in melanoma, and gleevec for *ABL-BCL* (translocation of *c-ABL* gene sequences from chromosome 9 into the *BCR* gene on chromosome 22) positive leukemia have been used widely in clinical settings [3]. However, cancer cells can evade death by gaining resistance to various treatment modalities, which has diluted the expected outcome of these therapies [4,5,6]. This has necessitated the discovery of alternative treatment strategies to cure cancer patients. Numerous in vitro studies in conjunction with ex vivo studies have exemplified the anti-cancer effects of natural products such as flavonoids [7,8,9,10]. Specifically, fisetin and quercetin, two well-studied flavonoids, have shown remarkable anti-cancer effects in multiple in vitro and in vivo systems. Different events in cancer initiation and progression such as apoptosis, extracellular matrix remodeling, epithelial to mesenchymal transition, cancer-associated inflammation, and oxidative stress can be controlled by fisetin and quercetin [11,12]. In vitro studies showed that fisetin and quercetin could also act against chemotherapeutic resistance in several cancers [13,14]. Numerous cancer-related molecules such as anti-apoptotic and pro-apoptotic proteins, cyclin-dependent kinases (CDKs), cyclins, matrix metalloproteinases (MMPs), and growth factors have been shown to be modulated by fisetin and quercetin (Figure 1 and Figure 2). Quercetin can also bind to a G protein coupled receptors to activate a G protein and calcium-dependent pathway which leads to tumor cell death [13,14,15]. In addition, these phytochemicals also exhibit synergistic effects where they enhance the anti-tumor activity of many anti-cancer drug molecules (Table 1). The present review highlights the important anti-cancer roles of quercetin and fisetin in various in-vitro/ex-vivo studies. 

## 2. Chemistry of Fisetin and Quercetin

Structurally, fisetin has two aromatic rings which are linked through a three-carbon oxygenated heterocyclic ring, and is supplemented with four hydroxyl group substitutions and one oxo group [38,39] (Figure 3a). It is usually present in different fruits and vegetables such as strawberries, apples, onions, and cucumbers [40] and in various trees and shrubs belonging to the Fabaceae and Anacardiaceae families, as well as *quebracho colorado* and pinophyta species [39]. Fisetin has low aqueous solubility and bioavailability. The biological activity of fisetin is due to the presence of hydroxyl groups at the 3, 7, 3’, 4’ positions and oxo group at the 4 position with double bond between C2 and C3. Quercetin belongs to the polyphenolic class and is found in many fruits, red onion, and the roots and leaves of many vegetables. It also has very poor solubility and oral bioavailability. Quercetin has five hydroxyl groups on C6-C3-C6 backbone structure, in particular a 3-OH group on the pyrone ring (Figure 3b) [41].

## 3. Regulation of Cancer-Related Processes

### 3.1. Activation of Intrinsic Apoptotic Pathway

Various cytotoxic chemotherapy drugs activate apoptotic processes in tumor cells by up-regulating and down-regulating the expression of different pro-apoptotic and anti-apoptotic proteins, respectively [42,43]. Similarly, flavonoids have been demonstrated to activate apoptotic processes in exorbitant cancer cell lines and animal models. Mechanistically, regulation of the release of cytochrome c from mitochondria, caspase-3 and caspase-9 mRNA and protein expression, and B-cell lymphoma 2 (Bcl-2) and Bcl-2 associated X (Bax) levels, were found to be regulated in the fisetin-treated cancer cell line (oral squamous carcinoma) [44]. Further, Li et al. also documented the apoptotic capacity of fisetin in T24 and EJ human bladder cancer cells, acting through overexpression of Bax, Bcl2 associated agonist of cell death (Bad), and Bcl-2 antagonist/Killer 1 (Bak) and inhibition of Bcl-2 and B-cell lymphoma-extra large (Bcl-xL) [45]. In a similar manner, apoptosis was seen in the U266 myeloma cancer cell line, which was mediated by activation of caspase-3, Bax, Bcl-2-like protein 11 (Bim), Bad, and inhibition of Bcl-2 and myeloid cell leukemia-1 (Mcl-1L) [46]. This phytochemical was also shown to exhibit an anti-tumor effect in the NCI-H460 NSCLC cell line through activation of caspase-9 and caspase-3 and inhibition of Bcl-2 and Bcl-xL, with adjacent effects on DNA fragmentation and depolarization of the mitochondrial membrane [47]. Another study identified that fisetin at 5–80 µM significantly reduced the viability of A431 human epidermoid carcinoma cells by the release of cytochrome c, reducing the anti-apoptotic protein expression of Bcl-2, Bcl-xL, and Mcl-1 along with elevation of pro-apoptotic protein expression (Bax, Bak, and Bad) and caspase cleavage and poly-ADP-ribose polymerase (PARP) protein [48]. The in-vitro and in-vivo administration of fisetin promoted caspase-8 and cytochrome c expression, possibly by impeding the aberrant activation of insulin growth factor receptor 1 and Akt proteins in oxaliplatin/irinotecan-resistant colorectal tumor cells [49]. In uveal melanoma cells, apoptosis was reported to be induced using fisetin by inhibiting the expression of Bcl-2 family proteins and increasing Bax levels, cytochrome c release, and activities of various caspases, whereas it was not cytotoxic to healthy retinal pigment epithelial cells [50]. The in vitro activity of fisetin in the mitochondrial apoptotic pathway has been acknowledged to be effective in the treatment of oral carcinomas. Fisetin enhanced the expression of pro-apoptotic proteins Bak and Bax but reduced anti-apoptotic protein expression (Bcl-2 and Bcl-x), while it also led to activation of caspase-3, caspase-8, caspase-9, and augmented sustained release of cytochrome c and apoptosis-inducing factor expression in HSC3 [51] and SCC-4 oral carcinoma cells [52]. Further, Khan et al. found fisetin (10-60 µM) treatment resulted in activation of apoptosis, poly (ADP-ribose) polymerase (PARP) cleavage, modulation of Bcl-2 family protein expression (Bak, Bad, Bid, Bcl-xL), inhibition of the phoshoinositide 3-kinase (PI3K)/Akt signaling pathway, and activation of caspase 3, caspase-9, and-8 enzyme activity in LNCaP prostate cancer cell lines [53]. Similarly, evidence suggests that quercetin can also activate apoptosis through a mitochondrial pathway involving the activation of caspase-3 and caspase-9 and by the release of cytochrome c and cleavage of PARP in acute lymphoblastic leukemia (HPB-ALL and HL-60) and prostate cancer cells (DU-145 and PC-3) [54,55,56]. Moreover, a myriad of studies has shown anti-apoptotic (Bcl-xL and Bcl-2) and pro-apoptotic (Bax) protein modulation by quercetin in human colon, oesophageal adenocarcinoma, and leukemia cells [57,58,59]. In addition, quercetin treatment also resulted in an attenuated Bcl-xL to Bcl-xS ratio and augmented translocation of Bax protein to mitochondrial membrane in LNCaP human prostate cancer cells [60]. In vitro studies with human cancer cell lines HaCaT keratinocytes, established the anti-tumorigenic effect of quercetin through Bax over-expression and release of cytochrome c and translocation of factors inducting apoptosis into the nucleus [61]. Similar effects of quercetin have also been reported in CLL and acute myeloid leukemia cell line HL-60; this phytochemical treatment activated the pro-apoptotic signaling cascade through PARP-1 cleavage and caspase activation, and also initiated autophagy events by the increased expression of light chain 3-II, decreased expression of p62, and formation of acidic vesicular organelles [62].

### 3.2. Activation of the Extrinsic Apoptotic Pathway

Several studies have illustrated the anti-tumor function of plant-derived products via activation of extrinsic apoptotic pathways [63,64]. Fisetin treatment administered in a time- and dose-dependent manner led to induction of apoptosis in HeLa cervical cancer by activation of caspases (3 and 8) and PARP cleavage [65]. Similarly, in human colon cancer cells (HCT-1160), fisetin-mediated apoptosis was also observed involving DNA condensation, cleavage of PARP, enhanced caspase-8 expression, Fas ligand, death receptor 5, and tumor necrotic factor-related apoptosis-inducing ligand (TRAIL-R1) expression [66]. Further, fisetin-promoted apoptotic activation was also seen in DU145, LNCaP, and PC3 human prostate cancer cells [53,67]. Quercetin (>20µM) induced caspase-dependent extrinsic apoptosis by upregulating the expression of caspase-3 and caspase-8, and inducing the cleavage of PARP in a HER2-overexpressing (BT-474) breast cancer cell line [68], consistent with the earlier reports in another leukemia cell line (CEM, K562 and Nalm6) [69].

### 3.3. Activation of Cell Cycle Arrest through the G0/G1 Check Point

Multiple in vitro studies have also noticed the cell cycle regulatory function of bioactive compounds mediated through CDKs and cyclin proteins [70,71]. Recent studies revealed that fisetin binds with CDK6, which in turn blocks its activity with an inhibitory concentration (IC_50_) at a concentration of 0.85 μM [72]. For instance, fisetin is identified as a regulator of cell cycle checkpoints, leading to cell arrest through CDK inhibition in HL60 cells and astrocyte cells over the G0/G1, S, and G2/M phases [73,74]. Sabarwal et al. and colleagues analyzed fisetin-induced proliferation inhibition in adenocarcinoma gastric cell line (AGS) and SNU-1 human gastric carcinoma cells, through a remarkable attenuation of G1 phase cyclins and CDKs level, while exhibiting elevated levels of p53 and its S15 phosphorylation [75]. Further, fisetin treatment has also been documented to arrest cell cycle growth in G0/G1 phase by enhancing the p53 and p21 gene expression, while reducing CDK4, CDK2, cyclin D1, and cyclin A in bladder cancer cell lines T24 and EJ [45]. In addition, at 10–60 μM fisetin concentration, prostate cancer cells PC3, LNCaP, and CWR22Ry1 had decreased cellular viability and decreased levels of D1, D2, and E cyclins and their activating partners CDK2, and CDKs 4/ 6, with consequent induction of KIP1/p27 and WAF1/p21 [53]. Fisetin (10–60 µM) treatment also shows a notable accumulation of the tumor cell population in the G1 phase of the cell cycle, accompanied by a concomitant decrease in the S-phase and G2/M-phase cell population [53]. 

Similar to fisetin, treatment of vascular smooth muscle cells with quercetin induced G1 cell cycle arrest through alleviation of D1/Cdk4 and E/Cdk2 and upregulation of p21 [76]. In addition, quercetin mediated anti-proliferation effects in different human cancer cells including hepatocellular carcinoma HepG2 cells, ovarian cancer SKOV 3 cells, and malignant mesothelioma (MM) MSTO-211H and H2452 cells [77] through cell cycle inhibition at G0/G1 to G2/M checkpoint [78]. In colon cancer cell lines, such as Caco2 [79] and SW480 [80], quercetin has been reported to inhibit cell cycle in a concentration-dependent (5 to 160 µM) manner. Furthermore, quercetin inhibited the cell cycle in colon cancer stem cells [81]. Quercetin-induced G0/G1-phase arrest occurred when expression of CDK2 and CDK4 was inhibited in HL-60 myeloid leukemia cells [62].

### 3.4. Activation of Cell Cycle Arrest through G2/M Check Point

Flavonoids can also block the cell cycle division at the G2/M check point [8]. For example, treatment of A431 cells with fisetin resulted in G2/M phase arrest [48]. Poor et al. demonstrated that among 18 tested compounds, diosmetin, fisetin, apigenin, luteolin, and quercetin provoked spectacular G2/M phase arrest in the hepatocellular carcinoma HepG2 cell line [82].

Similarly, quercetin enhanced the expression of retinoblastoma (Rb) gene in nasopharyngeal carcinoma cells HK1 and CNE2 and blocked the cell cycle in the G2/M phase [83]. Moreover, a concentration-dependent anti-tumor effect of quercetin was observed as a result of G2/M phase arrest in an in vitro study using lung carcinoma cell lines H1299 and A549A [84]. In addition, quercetin-mediated upregulation of p21, p27, p53, and Chk2 followed by downregulation of CDK1, cyclin B, pRb phosphorylation, and G2/M phase arrest were also reported in hepatocellular carcinoma [85,86]. Quercetin was also shown to inhibit cell cycle in G2/M phase in breast carcinoma (MCF-7) [87], leukemia (U937 cells) [88], and esophageal adenocarcinoma cell lines (OE33) [89].

### 3.5. Regulation of Extracellular Matrix Remodeling

The extracellular matrix (ECM) is an integral part of tissue and plays important functions [90]. Multiple studies described ECM remodeling as a key feature in lung, breast, ovarian, cervical, prostate, and colon cancer [91]. In a study, fisetin displayed tumor inhibitory effects by blocking MMP-2 and MMP-9 at mRNA and protein levels in prostate PC-3 cells [92]. Similarly, fisetin can also inhibit MMP-1, MMP-9, MMP-7, MMP-3, and MMP-14 gene expression linked with ECM remodeling in human umbilical vascular endothelial cells (HUVECs) and HT-1080 fibrosarcoma cells [93]. An interesting scientific finding from a recent study manifested that fisetin downregulates expression and reduces urokinase plasminogen activator (uPA) activity in human cervical adenocarcinoma SiHa and CaSki cells, responsible for activation of MMPs [94]. Furthermore, fisetin in a concentration-dependent manner (10–50 μM concentration) significantly inhibited regular serum, growth-enhancing supplement, and vascular endothelial growth factor (VEGF)-mediated growth in in vivo (mice) and in vitro (A549 and DU145, HUVECs) system, in addition to its effects on MMP-2 and MMP-9 [95].

A drop in MMP-2 and MMP-9 activity was detected in addition to its effects on other several apoptotic pathways after quercetin treatment in multiple cancer cell lines i.e., human head and neck squamous cell carcinoma (HNSCC), colon cancer (Caco-2 cells), and breast cancer (MCF-7 cells) [96,97,98]. Furthermore, anti-metastatic effects of quercetin were also explored and documented in the chicken chorioallantoic membrane assay using prostate cancer line PC-3 [99]. There also exists strong evidence that quercetin inhibits VEGF-related angiogenesis in several tumorigenic cell lines such as SN38 gastric cancer cells [100], osteosarcoma [101], and retinoblastoma (Rb) [102].

### 3.6. Regulation of Epithelial to Mesenchymal Transition

Epithelial to mesenchymal transition (EMT) is a key process in cancer invasion or progression [103,104,105,106,107,108,109]. Flavonoids have been shown to regulate the ECM remodeling and hence, cell invasion. The anti-metastasis effects of fisetin (5–20 μM) were revealed in melanoma cells, occurring through downregulation of mesenchymal markers (vimentin, N-cadherin, snail, and fibronectin) and upregulation of epithelial markers (desmoglein and E-cadherin) [110]. In in vitro and raft cultures, Pal et al. found that fisetin treatment minimized tumor invasion and tumor cell migration of BRAF V600E mutation-positive melanoma cells and alleviated EMT by decreasing vimentin, Twist1, N-cadherin, Snail1, ZEB1, Slug, and fibronectin expression, and escalating E-cadherin levels [16]. The in vitro findings from another study by Li et al. concluded that fisetin could significantly overpower growth and metastasis in MDA-MB-231 and BT549 triple-negative breast cancer cell lines, thereby blocking the EMT process induced through the phosphatase and tensin homolog/protein kinase B/glycogen synthase kinase 3 (PTEN/Akt/GSK-3β) signaling pathway [111].

Similarly, the recent scientific literature has also documented that quercetin prevented transforming growth factor beta (TGF-β)-induced EMT in PC-3 prostate cancer cells, through inhibition of TGF-β-induced expression of N-cadherin and vimentin along with increased E-cadherin expression. Additionally, quercetin significantly decreased the TGF-β-induced expression of Snail, Twist, and Slug [112]. Quercetin treatment was also illustrated to affect the migration capacity of head and neck cancer-derived sphere cells through decreased production of N-cadherin, twist, and vimentin [113]. Quercetin was also reported to decrease CT26 and MC38 colorectal carcinoma lung metastasis and also regulated the expression of EMT markers (E-cadherin, N-cadherin, Snail, and β-catenin) which in turn inhibited the expression of MMPs [114].

## 4. Control of Cancer-Associated Signaling Pathways by Fisetin and Quercetin

### 4.1. Regulation of PI3K/Akt Signaling Pathway

The PI3K/Akt pathway has been correlated with many fundamental processes during cancer development [115,116,117,118]. It plays a significant role in apoptosis, survival, and angiogenesis [119,120,121,122]. Therefore, this pathway could be a major target in cancer inhibition. Flavonoids have the potential to interact with this signaling pathway. Promising effects of fisetin on this pathway have been well reviewed by George [123]. Fisetin significantly reduced the viability of human osteosarcoma (U-2 OS) cells in a concentration-dependent manner (20–100 µm) through modulation of PI3K/Akt signaling cascades [124]. Similarly, fisetin inhibited PI3K expression and phosphorylation of Akt [53]. In a detailed study using fisetin, the blocking of Akt by exogenous siRNA in prostate cancer cells (LNCaP) caused an elevated Bad and Bax expression, and decreased expression of Bcl-2 and Bcl-xL, which suggested that these effects are mediated in part through Akt [53]. Further, in Raji cells (human Burkitt’s lymphoma cells), fisetin treatment activated the apoptotic process through inhibiting both PI3K and mammalian target of rapamycin (mTOR) signaling pathways [125]. Pal et al. presented the inhibitory effect of fisetin treatment on PI3K signaling pathway implicated through Akt phosphorylation of Akt in nude mice implanted with A375 and SKH-1 melanoma cells and SKH-1 hairless mice [16,126].

Moreover, a sustained inhibition of PI3K, Akt and cross-communication between PI3K and extracellular signal-regulated kinases (ERKs) was described in quercetin-treated HepG2 hepatocellular carcinoma cells [127]. An elevation in endocannabinoids receptor (CB1-R) expression following quercetin treatment has been noted in colon and breast cancer cell lines, which in turn inhibited survival signaling pathways such as PI3K/Akt/mTOR [128,129]. Cancer stem cells (CSCs) have recently gained major importance as novel targets in targeted cancer therapy and quercetin research is further supported by its inherent capacity to inhibit CSCs through the PI3K/Akt and MAPK/ERK pathways in prostate CSCs [130] and the PI3K/Akt/mTOR signaling pathways in breast CSCs [131].

### 4.2. Regulation of the Nuclear Factor Kappa Light Chain Enhancer of Activated B Cells Signaling Pathway

The nuclear factor kappa light chain enhancer of activated B cells (NF-κB) signaling pathway has a remarkable role in pathologic conditions such as cancer [132,133]. Suppression of NF-κB pathway in tumorigenic cells usually leads to tumor regression, which makes the NF-κB pathway a promising therapeutic target [134,135]. The effects of fisetin on this pathway have been investigated by Sung et al. in several cancer cell lines including Daudi human Burkitt lymphoma cells, H1299 human lung adenocarcinoma cells, and A293 human embryonic kidney cells. Fisetin inhibited TNF-induced IκBα degradation, NF-dependent IκBα phosphorylation and ubiquitination, TNF-induced IκBα kinase activation, p65 phosphorylation and its nuclear translocation, NF-κB-dependent anti-apoptotic gene expression (cIAP1/2, survivin, Bcl-2, XIAP, Bcl-xL and TRAF-1), and expression of cyclin D1, c-Myc, COX-2, MMP-9, VEGF, and intercellular adhesion molecule-1 (ICAM-1), thereby revealing the numerous effects of fisetin on this pathway [17]. In another study, the treatment of COX-2 overexpressing HT29 human colon cancer cells with fisetin resulted in activation of apoptosis and inhibition of COX-2 and the Wnt/EGFR/NF-kB pathway [136]. In addition, fisetin treatment reversed 12-O-tetradecanoylphorbol-13-acetate (TPA) mediated cell migration in MCF-7 human breast cancer cells, which caused NF-κB inactivation and downregulation of MMP-9 expression [137]. Another study found fisetin mediated downregulation of Syk, Src, and IκBα through inhibition of nuclear translocation of p65/NF-κB [138]. 

Youn et. al. confirmed that quercetin inhibited the growth of NSCLC (H460) by suppressing the NF-κB [139]. Another study, investigated the time dependent inactivation of NF-κB pathway with NF-κB binding activity which leads to the reduced survival and proliferation of HepG2 cell [140]. In human cell colon cancer (CACO-2 and SW-620 cell) quercetin also known to block the proliferation via inhibition of NF-κB pathway [141]. 

### 4.3. Regulation of JAK/STAT Signaling Pathway

The effects of Janus kinase/signal transducers and activators of transcription (JAK/STAT) signaling on tumor cell survival, proliferation, and invasion have made this pathway a favorite target for drug development and cancer therapy [142]. It was found that fisetin mediated apoptosis by regulation of the JAK/STAT and c-Kit pathways in K562 human chronic myeloid leukemia cells [143].

Additionally, it was suggested that quercetin inhibited the transcriptional activity of STAT3 and reduced the expression of STAT3 targeted genes such MM 2, MMP 9, Mcl-1, and VEGF in melanoma [144]. Furthermore, treatment with quercetin at 10 μM markedly inhibited JAK2 and STAT1 phosphorylation, and nuclear translocation of phosphorylated STAT1 in poly (dA:dT)-treated and interferon gamma-primed keratinocytes [145]. Michaud et al. suggested that quercetin reduced interleukin-6 (IL-6), stimulated JAK1 and STAT3 activation, and subsequently reduced the recruitment of cyclin D1 and MMP 2 genes in glioblastoma [146,147]. Mukherjee et al., described that quercetin also showed anti-tumor effects through downregulating the IL-6/STAT3 signaling pathway in NSCLC A549 cells [148]. 

### 4.4. Regulation of p38MAPK Pathway

Deregulation of p38 MAPK has been associated with advanced stages and short survival in cancer patients [149]. In human NCI-H460 NSCLC, fisetin regulated production of reactive oxygen species (ROS) and reduced the activation of p38 MAPK signaling pathway [150]. Similarly, the fistein treatment in HL-60 human acute promyelocytic leukemia cells caused inhibition of MAPK signaling and modulated DNA binding signaling pathways [73]. Furthermore, fistein treatment was also documented to reduce TPA-induced MCF-7 breast cancer cell invasion through inactivation of p38 MAPK signaling [137]. 

It was found that quercetin-mediated apoptosis was induced by ROS-dependent ASK1⁄p38 pathway activation [151]. In AGS gastric cancer cells, quercetin inhibited the MAPK cancer associated pathway and TRPM7 channels thereby acting as potential therapeutic agent for gastric carcinoma treatment [152]. Quercetin additionally induced apoptosis in breast cancer cells (MCF-7 and MDA-MB-231) through inhibition of p38 MAPK signaling leading to decrease in twist gene expression [153]. Lim et al. suggested that quercetin inhibited the phosphorylation of Akt, P70S6K, and S6 proteins, while it increased phosphorylation of P38, c-Jun N-terminal kinase (JNK), ERK1/2, and P90RSK proteins in JAR and JEG3 choriocarcinoma cell lines [154].

### 4.5. Regulation of ERK Signaling Pathway

The ERK signaling pathway is another major determinant for diverse cellular processes such as proliferation, differentiation, and survival [155]. In addition, various in vitro and ex vivo studies have also determined the anti-tumor effects of flavonoids mediated through ERK1/2 signaling inhibition. Fisetin is one of the flavonoids that has been found to suppress ERK1/2 signaling in human gastric (SGC7901), hepatic (HepG2), colorectal (Caco-2), and pancreatic cancer cells (Suit-2) [156,157]. In human NCI-H460 NSCLC, fisetin induced ROS generation and suppressed ERK through its phosphorylation [150]. Fisetin has also been documented to decrease the survival in CCA cholangiocarcinoma cells through modulating ERK phosphorylation, reduction of phospho-p65 and c-myc oncogene expression [158]. Similarly, a combination of *N*-acetylcysteine treatment and fisetin inhibited ERK protein phosphorylation in COLO-205 colon carcinoma cells [159]. Another study also found that fisetin had an anti-tumor effect in human glioma GBM8401 cells, via the regulation of ERK 1/2 and ADAM9 expression [160]. Further, quercetin caused HL-60 myeloid leukemia cell death by ERK signaling mediated apoptosis [161]. Quercetin was found to be engaged in inhibiting the ERK pathway, which subsequently suppressed angiogenesis [162,163]. Furthermore, recent data revealed that quercetin decreased prostate CSC survival and invasion through regulating MAPK/ERK signaling pathways [130].

### 4.6. Regulation of the Akt/mTOR/p70S6K Pathway

A range of studies has described the PI3K/Akt/mTOR pathway’s involvement in the initiation of angiogenesis. Natural products such as flavonoids may be used to target this cancer-related pathway [115]. Fisetin reduced Akt phosphorylation, p70S6K, mTOR, and mitf proteins in 451Lu human melanoma cells, which in turn inhibited angiogenesis [164]. Furthermore, in addition to melanoma [18], fisetin was found to be effective in leukemia [165] and lymphoma cells through this pathway. Following treatment with fisetin, the viability of 4T1, MCF-7, and MDA-MB-231 breast cancer cells was reduced through interfering with the PI3K/Akt/mTOR pathway [166]. Fisetin was found to be an inhibitor of PI3K/Akt/ mTOR pathways [167] and an inducer of autophagia [168] in prostate cancer cell lines.

Quercetin was also documented to impede tumor proliferation and angiogenesis by targeting the VEGF receptor-2 (VEGF-R2) and Akt/mTOR/P70S6K signaling pathway in a mouse prostate cancer xenograft model [169]. The Akt/AMPK/mTOR pathway was shown to be one of the targets of quercetin in MDA-MB-231 and MDA-MB-435 breast cancer cells [170]. Furthermore, quercetin also induced Akt-mTOR mediated autophagy in breast cancer cells where it reduced the migration and metastasis of cells through MMP-2, MMP-9, and VEGF inhibition [171].

### 4.7. Regulation of Nrf2 Signaling Pathways

The Kelch-like ECH-associated protein 1 (Keap1)-nuclear factor E2-related factor 2 (Nrf2) pathway is the major signaling cascade for defense and survival against endogenous and exogenous stress [172,173]. It has been found that fisetin increased the protein level and accumulation Nrf2 and down regulated the protein levels of Keap1 [174]. Fisetin also rapidly enhanced the expression of both Nrf2 and activating transcription factor 4 (ATF4) along with distinct mechanism leading to ATF-dependent gene transcription [175]. Treatment of cancer cells with quercetin caused dissociation of the Nrf2-Keap1 complex, and translocation of Nrf2 to the nucleus [176,177]. It has been found that treatment of HepG2 hepatocellular carcinoma cells with quercetin initiated antioxidant response element (ARE) binding activity and NQO1 expression [177]. Additionally, quercetin also prevented degradation of Nrf2 and stabilized it by maintaining the Keap1 protein levels without affecting Keap1-Nrf2 complex [177]. Further, quercetin-mediated overexpression of detoxification enzymes (phase II) was also demonstrated in human HuTu 80 duodenum adenocarcinoma cells and Caco-2 colorectal adenocarcinoma cells [178]. Recently, the anti-tumorigenic effects of quercetin was reported to be mediated by increased Nrf2 expression in MSTO-211H and H2452 malignant mesothelioma cells [179].

## 5. Regulation of Various Growth Factors

Growth factors are the key mediators for activation of several cancer-related signaling pathways [180]. Growth factors influence cell transformation and activation of growth-promoting pathways in tumor cells [181]. Flavonoid compounds are also able to control the expression of these growth factors. Mechanistically fisetin-induced anti-angiogenesis led to reduced VEGF and epidermal growth factor receptor (EGFR) expression [76,136]. Several in vivo studies in rats narrated downregulation of bVEGF and basic fibroblast growth factor which direct tumor growth inhibition, and alleviation of tumor proliferation, and angiogenesis [19]. Quercetin attenuated tumor proliferation, invasion, and migration through inhibiting hepatocyte growth factor (HGF)/c-Met signaling in melanoma cells [182]. Quercetin is also reported to reduce the insulin-like growth factors (IGFs) via increasing binding protein-3 (IGFBP-3) proteins and led apoptosis in the prostate cancer cell line PC-3 [183]. The effects of quercetin was promising enough to formulate novel gold nanoparticle conjugates of quercetin that induced apoptosis [184] and inhibited EMT, angiogenesis, and invasiveness through EGFR/PI3K/Akt and EGFR/VEGFR-2-mediated pathways respectively [185].

## 6. Regulation of Pro-Inflammatory Cytokines

Inflammation has been correlated with 20–40% of cancer cases. The blocking of cytokines is best used as an adjunct therapy together with tumoricidal drugs [186]. Fisetin attenuated aflatoxin B1 (AFB1)-mediated carcinogenesis by neutralizing elevated levels of IL-1α and TNF-α in rat model of hepatocellular carcinoma [187]. Other research groups have demonstrated a fisetin-mediated inhibitory effect against pro-inflammatory cytokines (e.g., TNF-R, IL-6, IL-8, and IL-1β), nitric oxide, and Th2-type cytokines (basophil-induced) in human mast cells [188,189]. The key allergic airway inflammation mediators, including Th2-associated cytokines (IL-13, IL-4, and IL-5), thymic stromal lymphoprotein, eotaxin-1, and transcription factor (GATA-3) in lungs respectively, were known to have reduced expression after fisetin treatment [190]. Moreover, fisetin is also known to reduce the level of inflammatory cytokines (IL-6, TNFα, and IL-1βIL-6) and expression of cell proliferation markers [126]. Fisetin suppressed IL-1β-mediated expression of inducible nitric oxide synthase, nitric oxide, interleukin-6, tumor necrotic factor-α, prostaglandin E2, cyclooxygenase-2 (iNOS, NO, IL-6, TNF-α, PGE2, and COX-2), and significantly decreased the degradation of Sox-9 and aggrecan, and reduced SIRT1 inactivation. In contrast, quercetin supplementation significantly decreased the infiltration of inflammatory cells as well as the levels of TNF-α and IL-1β in the bronchoalveolar lavage fluid and plasma of gerbils exposed to benzo[a]pyrene (BaP) or BaP+ β-carotene in A549 adenocarcinoma alveolar basal epithelial cells [191]. Quercetin inhibited TNF-induced interferon-γ-inducible protein 10 and macrophage inflammatory protein 2 gene expression in MODE-K cells [192]. Quercetin attenuated IL-1β-induced expression of ICAM-1 mRNA and protein in a dose-dependent manner in human A549 adenocarcinoma alveolar basal epithelial cells [193].

## 7. Regulation of Heat Shock Proteins

Heat shock proteins (HSPs) such as HSP70, HSP90, and HSP27 stabilize the functions of overexpressed and mutated cancer genes, and hence, increase the growth and survival of cancers [194]. HSPs are also released from cancer cells and influence malignant properties by receptor-mediated signaling [195]. Researchers reported that quercetin stimulate apoptosis in prostate cancer and B-CPAP human papillary thyroid cancer cells resulting from inhibition of casein kinase II or calcium/calmodulin kinase II, leading to HSP induction and downregulation of HSP70 and HSP90 protein expression [196]. Quercetin found to suppressed the HSP27 and COX-2 and induced G1 phase arrest in U251 glioma cells [197]. Numerous studies have documented the promising role of HSP90 protein in combating cancer, as this supreme protein molecule is reported to be involved in cell survival pathways [20,198,199]. Kim et al. presented and discussed the fisetin-mediated inhibition of cellular proliferation by HSP70 and HSP27 regulation, and Bcl-2-associated athanogene 3, which can stabilize Bcl-2 protein family members, thereby protecting cancer cells from apoptosis in HCT-116 human colon cancer cells [200]. One of the paramount signaling pathway, Notch/AKT/mTOR in tumor aggressiveness, has reported to be downregulated by quercetin plus shHSP27, leading to significant apoptosis in U937 leukemia cells [201]. 

## 8. Conclusions and Future Perspectives

Fisetin and quercetin are some of the most prevalent plant flavonoids that are reportedly present in many fruits and vegetables such as apples and onions. The bioactive potential of fisetin and quercetin has been established, especially in the modulation of a range of cancer signaling pathways. The anti-cancer, anti-inflammatory, and antioxidant roles exhibited by these flavonols have been reportedly found to be associated with their ability of apoptotic activation, cell cycle arrest, regulating ECM remodeling, and inhibiting EMT. Many studies of fisetin and quercetin have been highlighted in this review for their modulatory potential in different cancer related signaling pathways and growth factors such as Akt, JNK, p38MAPK, NF-κB, and VEGF cytokines and chemokines, paving the way to delineate the mechanism of action of these therapeutically active flavonols. Research is in progress to identify the newer plant based therapeutic agents. Easley availability, safe in use and cheaper cost increasing the popularity and acceptability of herbal medicine. As per the World Health Organization, about 60% of the world´s population and about 80% of the population of developing countries rely on herbal medicine. According to an estimate, herbal industry reached at around US$100 billion in shares, with an annual growth rate of about 15%. However, there are also several concerns with use of these phytochemicals as drugs with respect to their pharmacognosy and standardization compared with conventional drugs. There should be an improvement in the technologies used for the categorization (such as HPLC/MS, LC/MS, NMR), storage, and quality control of these compounds. For the last two decades, research has been underway to evaluate the clinical validity of phototherapy. Clinical trials are required to further prove the clinical efficacy of these phytochemicals. 

## Figures and Tables

**Figure 1 biomolecules-09-00174-f001:**
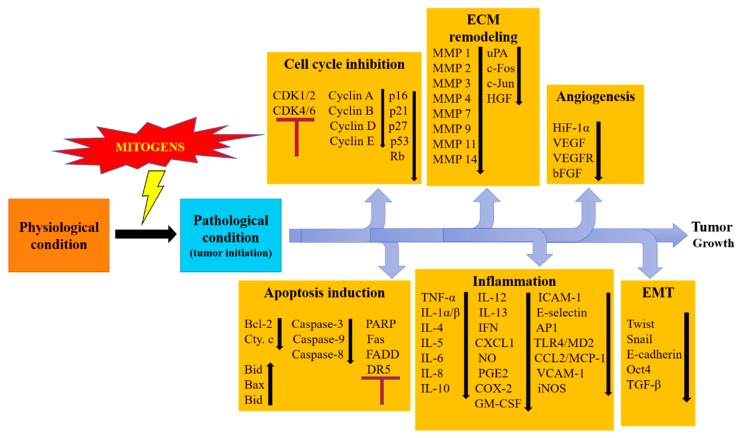
Showing regulation of different cancer related processes under the effect of flavonoids. All these processes are crucial in carcinogenesis and play key roles for cancer initiation and progression. Flavonoids effectively act at these processes to inhibit cancer growth.

**Figure 2 biomolecules-09-00174-f002:**
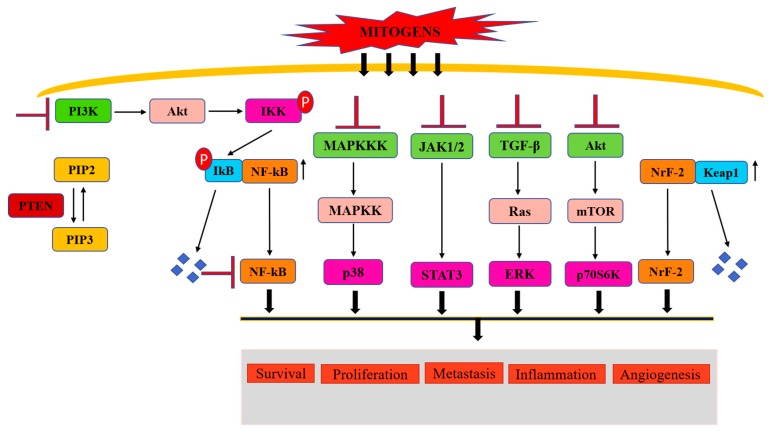
Figure showing regulation of different cancer associated signaling pathways. Deregulation of these pathways has been determined in human malignancies. Flavonoids control the deregulation of these pathways and cancer proliferation.

**Figure 3 biomolecules-09-00174-f003:**
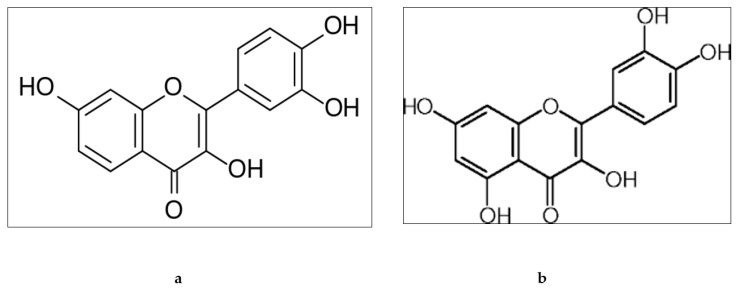
Chemical structure of fisetin (**a**) and quercetin (**b**).

**Table 1 biomolecules-09-00174-t001:** Synergistic effects.

Phytochemical	Adjunctive Drug	Mechanism	Model	Reference
Fisetin	Cisplatin	Inhibits the MAPK signaling pathway and downregulates survival proteins	A549-CR	[12]
	Sorafenib	Anti-invasive and anti-metastatic	A375 and SK-MEL-28	[16]
	doxorubicin	Potentiates the cytotoxicity of cisplatin	H1299	[17]
	Sorafenib	Downregulates the MAPK and PI3K pathways	*B-Raf*-mutated melanoma cells	[18]
	Cisplatin	Activates intrinsic and extrinsic apoptosis pathways	NT2/D1	[19]
	Geldanamycin	Activates intrinsic apoptosis pathways	COLO-205	[20]
	Paclitaxel	Autophagic cell death	A549	[21]
	Cisplatin	Cytotoxic	Rat model	[22]
	Etoposide	Cytotoxic	Saos-2	[23]
	Cyclophosphamide	Anti-angiogenic effect	Mice xenograft	[24]
	Sorafenib	Activates intrinsic and extrinsic apoptosis pathways	HeLa cells and HeLa xenograft	[25]
	Luteolin	Cytotoxic	HG-3 and EHEB	[26]
Quercetin	EGCG	Suppresses the JAK/STAT cascade	CCA cells	[27]
	Sulforaphane	miR-let7-a mediated inhibition of *K-ras*	PDA	[28]
	Methoxyestradiol	Apoptosis and G2/M phase arrest	LNCaP and PC-3 cells	[29]
	Cisplatin and Oxaliplatin	Cytotoxic	Ovarian tumor model	[30]
	Cisplatin	Modulates the miR-217–*K-ras* axis	143B cells	[31]
	Renistein	Modulates expression of androgen receptors and NQO1	CWR22Rv1 cells	[32]
	Imperatorin	Apoptosis	T98G	[33]
	Resveratrol	Modulates metabolic pathways	Adipose tissue triacylglycerol	[34]
	Doxorubucin	G2/M cell cycle arrest	HT29 cell	[35]
	Cyclophosphamide	Cytotoxic	Bladder cancer patients	[36]
	Cisplastin	Cytotoxic	SPC212 and SPC111 cell line	[37]

MAPK: Mitogen-activated protein kinase, PI3K: Phosphoinositide 3-kinase, JAK: Janus kinase, STAT: Signal transducer and activator of transcription, NQO1: NAD(P)H Quinone Dehydrogenase 1, EGCG: Epigallocatechin gallate, A549-CR: lung adenocarcinoma cispltin resistant, SK-MEL-28: Skin-malignant melanoma, H1299: Human non-small cell lung carcinoma cell line, NT2/D1: Pluripotent human testicular embryonal carcinoma cell line, COLO-205: Human colorectal adenocarcinoma cell line, Saos-2: Sarcoma osteogenic, HeLa: Human cervical cancer cells, HG-3/ EHEB: Chronic lymphocytic leukemia, CCA: Cholangiocarcinoma, PDA: Pancreatic ductal adenocarcinoma, LNCaP/PC-3: Human prostate adenocarcinoma cells, CWR22Rv1: Prostate cancer cell line, T98G: Human brain glioblastoma, HT-29: Colorectal adenocarcinoma, SPC212/ SPC111: Pleural biphasic mesothelioma.

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
