# Peer review of "Fisetin and Quercetin: Promising Flavonoids with Chemopreventive Potential"

_biomolecules, 2019, doi:10.3390/biom9050174_

Round 1

Reviewer 1 Report

The authors of the manucript "Fisetin And Quercetin: Promising Flavonoids With Chemopreventive Potential" reported the chemopreventive potential of Fisetin and Quercetin.

The review is well written and pharmacologically well organized.

There are some mistakes in the introduction, starting from line 58 to line 64 there is a confusion in the used words. It is not clear how fisetin and quercetin have been chosen to write this review.

Please, check this point accurately and control the word "in vitro" in all the manuscript. It is written in more ways.

In the main body of the paper, there is a rigth harmony of contents from my point of view.

It is also interesting if you add in the manuscript for example, the involvement of G-protein coupled receptors in the chemopreventive activity of these two flavonoids.

Author Response

Author(s) would like to appreciate the contributions of all the reviewers and editor for reviewing this manuscript. Author(s) accepted and appreciating all the valuable comments made by the reviewers. These comments really improve the quality of work of the manuscript and definitely help the readers. 

Reviewer 2 Report

This review summarized the anti-tumor effects and classified the mechanisms of fisetin and quercetin. However, the manuscript is not well written and organized. 

1, some of the classification is not precise. Some of the reviewed studies show contrast results (e.g. quercetin and STAT1; fisetin and Nrf2), and the authors have not given explanations.

2, many studies that relate to the topic are not included and many irrelevant studies are cited (e.g. quercetin and NF-kB).

3, many abbreviations are not defined or shown in order. Some of them have been abbreviated incorrectly (e.g. protein kinase B (AKT); nuclear factor kappa B activated cells (NF-κB) ).

4, the figures are not presented well. Because not all the effects are mediated via "Blocking".

5, where is "Table 1"?

6, some cell lines are not correct (e.g. Nalm6 is B cell precursor leukemia cell line but not breast cancer cell line).

7, some of the sentences are missing (page 2 line 60-63). 

A real critical dissection and discussion of the literature is missing. In view of the missing innovativeness of the paper, I do not give priority for this review to be published.

Author Response

(The authors gave the same response as above.)

Round 2

Reviewer 1 Report

The revised version of the manuscript highlights several new points, confirming the value of this review.

Author Response

Rebuttal Letter (Round second)

Manuscript ID: biomolecules-487395

Type of manuscript: Review

Title: Fisetin And Quercetin: Promising Flavonoids With Chemopreventive Potential

Comment by author(s): Author(s) would like to appreciate the contributions of all the reviewers and editor for reviewing this manuscript. Author(s) accepted and appreciating all the valuable comments made by the reviewers. These comments really improve the quality of work of the manuscript and definitely help the readers.

Hereby author(s) submitting the answers for each comments.

1.      Comments and Suggestions for Authors

The revised version of the manuscript highlights several new points, confirming the value of this review.

Response: Author(s) are thankful for appreciating the manuscript.

2.      Comments and Suggestions for Authors

Please address the following issues.

Comment-1: Please include a brief introduction of fisetin and quercetin. 

Response: A paragraph with heading “Chemistry of fisetin and quercetin” is added in the manuscript. This paragraph comprising chemical and structural properties of fisetin and quercetin.

Comment-2a: Please add "future perspective" in Section 7. As it now only appears to be a Conclusion. 

Response: Section 7 is revised as suggested by the reviewer.

Comment-2b: Also, as a reader, I am interested in the differences in the structures/mechanism between these two flavonoids with high structural similarity.

Response: A new paragraph “ Section 2” is added in the manuscript. This particular section explaining the differences in the structures/mechanism of these two flavonoids

Comment-3: Please make sure to cite proper references and remove any irrelevant references, as it's quite crucial in a scientific paper. Please check & modify thoroughly. Some of the examples include [1-4] of line 39 (can these Refs support the description ''The incidence of malignant diseases and the prevalence of cancer mortality is proliferating at an amplified rate across the developed and developing countries"?); [11-34] of line 55; [22, 23, 40-48] of line 64; Ref [70] of line 113 has nothing to do with the reviewed contents; [79] of line 133; [116, 117, 125-131] of line 223 (the Refs here should be those which describe the involvement of PI3K/Akt pathway during cancer development but not certain pharmacology study of non-flavonoids natural products.) [142-158] of line 247; [33, 144, 149, 152, 159-168] of line 249; [176-191] of line 276; [247] of line 394; etc.

Response: All these suggestions of the reviewer’s are undertaken and irrelevant references are removed from the text.

Comment-4: Immunotherapy should also be included in line 46.

Response: Word “ Immunotherapy” is added in suggested line/sentence.

Comment-5: inhibit cell cycle arrest (line 153)?; TGF-β from line 204-206

Response: Both these mistakes are rectified.

Reviewer 2 Report

Please address the following issues.

Please include a brief introduction of fisetin and quercetin. 

Please add "future perspective" in Section 7. As it now only appears to be a Conclusion. Also, as a reader, I am interested in the differences in the structures/mechanism between these two flavonoids with high structural similarity. 

Please make sure to cite proper references and remove any irrelevant references, as it's quite crucial in a scientific paper. Please check & modify thoroughly. Some of the examples include [1-4] of line 39 (can these Refs support the description ''The incidence of malignant diseases and the prevalence of cancer mortality is proliferating at an amplified rate across the developed and developing countries"?); [11-34] of line 55; [22, 23, 40-48] of line 64; Ref [70] of line 113 has nothing to do with the reviewed contents; [79] of line 133; [116, 117, 125-131] of line 223 (the Refs here should be those which describe the involvement of PI3K/Akt pathway during cancer development but not certain pharmacology study of non-flavonoids natural products.) [142-158] of line 247; [33, 144, 149, 152, 159-168] of line 249; [176-191] of line 276; [247] of line 394; etc.

Immunotherapy should also be included in line 46.

inhibit cell cycle arrest (line 153)?; TGF-β from line 204-206;   

Author Response

(The authors gave the same response as above.)

Round 3

Reviewer 2 Report

Thank you for the revisions. 

In the Abstract, "Despite advancements in healthcare facilities for diagnosis and treatment, cancer still endures to be the primary cause for worldwide mortality", cancer is not the primary cause of death, heart disease is. The author may consider to rewrite it. 

Author Response

Rebuttal Letter (Third second)

Manuscript ID: biomolecules-487395

Type of manuscript: Review

Title: Fisetin And Quercetin: Promising Flavonoids With Chemopreventive Potential

Comment by author(s): Author(s) would like to appreciate the contributions of all the reviewers and editor for reviewing this manuscript. Author(s) accepted and appreciating all the valuable comments made by the reviewers. These comments really improve the quality of work of the manuscript and definitely help the readers.

Hereby author(s) submitting the answers for each comments.

Comments and Suggestions for Authors

Comment: In the Abstract, "Despite advancements in healthcare facilities for diagnosis and treatment, cancer still endures to be the primary cause for worldwide mortality", cancer is not the primary cause of death, heart disease is. The author may consider to rewrite it. 

Response: Author(s) are agreed with the reviewer’s comment. As suggested, this particular line is modified.